biophysics/cellular biophysics

quasi-elastic neutron scattering, red blood cells, protein diffusion, small-angle neutron scattering, protein interactions, haemoglobin

**Authors for correspondence:**
Christopher J. Garvey
e-mail: christopher.garvey@mau.se
Andreas M. Stadler
e-mail: a.stadler@fz-juelich.de

[†]These authors contributed equally to this study.
[‡]Present address: ISIS Pulsed Neutron and Muon Facility, Science and Technology Facilities Council, Rutherford Appleton Laboratory, Harwell Science and Innovation Campus, Oxon OX11 0QX, UK.

# Effect of red blood cell shape changes on haemoglobin interactions and dynamics: a neutron scattering study

Keyun Shou[1,2,3,†], Mona Sarter[1,4,†,‡],
Nicolas R. de Souza[3], Liliana de Campo[3],
Andrew E. Whitten[3], Philip W. Kuchel[5],
Christopher J. Garvey[3,6,7] and Andreas M. Stadler[1,2]

[1]Jülich Centre for Neutron Science (JCNS-1) and Institute of Biological Information Processing (IBI-8: Neutron Scattering and Biological Matter), Forschungszentrum Jülich GmbH, 52428 Jülich, Germany
[2]Institute of Physical Chemistry, RWTH Aachen University, Landoltweg 2, 52056 Aachen, Germany
[3]Australian Nuclear Science and Technology Organisation, Lucas Heights, New South Wales 2234, Australia
[4]I. Physikalisches Institut (IA), AG Biophysik, RWTH Aachen, Sommerfeldstrasse 14, 52074 Aachen, Germany
[5]School of Life and Environmental Sciences, University of Sydney, Sydney, New South Wales, Australia
[6]Biofilm—Research Center for Biointerfaces and Biomedical Science Department, Faculty of Health and Society, Malmö University, Malmö, Sweden
[7]Lund Institute for Advanced Neutron and X-ray Science, Lund, Sweden

MS, 0000-0003-1867-5543; NRdS, 0000-0003-4099-9544;
LdS, 0000-0003-4799-2935; AEW, 0000-0001-8856-3120;
PWK, 0000-0003-4100-7332; CJG, 0000-0001-6496-7008;
AMS, 0000-0003-2272-5232

By using a combination of experimental neutron scattering techniques, it is possible to obtain a statistical perspective on red blood cell (RBC) shape in suspensions, and the inter-relationship with protein interactions and dynamics inside the confinement of the cell membrane. In this study, we examined the ultrastructure of RBC and protein–protein interactions of haemoglobin (Hb) in them using ultra-small-angle neutron scattering and small-angle neutron scattering (SANS). In addition, we used the neutron backscattering method to access Hb motion on the ns time scale and Å length scale. Quasi-elastic neutron scattering (QENS) experiments were performed to measure diffusive motion of Hb in RBCs and in an RBC lysate. By using QENS, we probed both internal Hb dynamics and global protein

diffusion, on the accessible time scale and length scale by QENS. Shape changes of RBCs and variation of intracellular Hb concentration were induced by addition of the Na$^+$-selective ionophore monensin and the K$^+$-selective one, valinomycin. The experimental SANS and QENS results are discussed within the framework of crowded protein solutions, where free motion of Hb is obstructed by mutual interactions.

## 1. Introduction

Haemoglobin (Hb) is the major macromolecular fraction of red blood cells (RBCs). In solution, the Hb protein is a homotetramer that consists of two α- and two β-chains bearing in total four haem groups [1]. Only higher vertebrates produce highly specialized RBCs for oxygen transport in their cardiovascular circulations. The Hb concentration in RBCs is high, with a value $c = 330$ mg ml$^{-1}$ and a volume fraction, $\phi \cong 0.25$, under standard physiological conditions [2]. The high intracellular Hb concentration has a strong impact on several cellular properties, on the molecular scale. Due to the high volume-fraction of Hb, many other solutes are excluded from a large fraction of the cell's cytoplasm, a phenomenon referred to as 'macromolecular crowding'. This effect significantly increases protein folding stability, and enhances (bio)chemical reaction rates, while altering equilibrium constants of (bio)chemical reactions [3]. Molecular crowding in RBCs modifies Hb–Hb interactions as reflected in a reduced rate of diffusional mobility [2,4–7]. Hb is constrained by the plasma membrane to the cytoplasm of the RBC. RBC shape (the biconcave disc or discocyte) and volume are regulated by, at least, the Na,K-ATPase [8], the calcium activated K-channel (Gárdos channel, KCa$_{3.1}$; KCNN4), and the mechanosensitive cation channel Piezo1, working in concert with Ca-ATPase [9]. Hence, the consumption of glucose via the glycolytic pathway is imperative for the phosphorylation of ADP to generate ATP that drives the two active pump systems. During RBC ageing, glucose consumption declines and RBC shape is not maintained, leading first to echinocytes (spiculated spheres) and then via loss of membrane vesicles to spherocytes (smooth spheres) [10,11]. RBC morphology can also be altered osmotically by increasing the K$^+$ and Na$^+$ concentrations in the cytoplasm, leading to swelling and dilution of the Hb therein; whereas the Hb concentration is increased in the shrunken spherocytes.

Previous studies have suggested that Hb diffusion in discocytes has been 'optimized by evolution', thus maximizing the oxygen uptake rate in the lungs, and release of oxygen in the tissues [6]. However, this is only part of the explanation for the evolved RBC shape, because there is also a geometrical constraint: RBC must undergo substantial deformation as they pass through capillaries with diameters that are as small as a half of their main diameter and yet on the time scale of their transit (approx. 300 ms in the lungs [9]) both cell volume and surface area are maintained despite the reversible deformation. In other words, the dimples of the discocyte enable the cell's extreme flexibility. Therefore, investigations on the bulk flow properties of concentrated Hb solutions and Hb diffusion rates that occur in response to RBC shape changes are needed to provide an understanding of the operation of RBCs in the whole body, at the molecular level.

Neutron scattering is a robust method, in terms of its applicability to highly concentrated solutions that strongly absorb photons (in other words, with solutions that are optically opaque) [6,7,12,13]. Furthermore, neutron scattering techniques do not induce any radiation damage, so they are well suited to investigating metabolically active biological samples *in vivo*, or when applied to concentrated protein solutions *in vitro* [14]. Small-angle neutron scattering (SANS) enables the investigation of protein structures on the nanometre length scale; it is a sensitive probe of protein–protein interactions that arise at high protein concentrations, such as in RBCs. Ultra-small-angle neutron scattering (USANS) is a unique technique that allows access to even smaller scattering angles, corresponding to larger length scales that extend into the micrometre length range [15]. Hence, the combination of USANS and SANS enabled us to investigate the links between intracellular Hb–Hb interactions and RBC shape; in other words, we could investigate even transient protein–protein structures from nanometres to several micrometres (in reciprocal space). On the other hand, neutron spectroscopy probes motions in biological matter, at thermal equilibrium. A limitation of quasi-elastic neutron scattering (QENS) is the restricted time scale and length scale over which it applies to measure solute movement. QENS is sensitive in the very short (fast) time scales of the order of nanosecond, and atomic length scale of the order of angstroms to nanometre. Incoherent neutron scattering, such as is probed by high-resolution QENS, provides direct information on global protein

self-diffusion in terms of the self-diffusion coefficient $D_{self}$, as well as internal protein dynamics. It has been a valuable tool in studies of protein motion on the nm–Å length scale in cells and small organisms [16–21].

It transpires that USANS/SANS and QENS are directly applicable, with minimal sample preparation, to investigating how the shape of an RBC modifies Hb–Hb interactions that alter the self-diffusion coefficient of Hb in the concentrated solution of the cytoplasm, *in situ*. Samples irradiated by the neutron beam are typically millilitres in volume, so the experimental perspective is truly statistical, given the billions of RBCs present.

Overall, we report on the use of USANS/SANS and high-resolution QENS experiments to probe the effect of RBC shape changes on Hb–Hb interactions. We studied the dependence of $D_{self}$ of Hb in RBCs under physiological conditions and in cells whose shapes were altered by the $Na^+$ ionophore monensin and the $K^+$ ionophore valinomycin.

# 2. Material and methods

## 2.1. Sample preparation

Fresh blood was collected from domestic horses at the University of Sydney's Camden Equine Centre and from healthy human volunteers (CJG or PWK) by venepuncture under University of Sydney, Human Ethics Clearances, and anticoagulated with 60 µl of 5000 IU ml$^{-1}$ heparin. While the healthy horse RBC has a slightly smaller average volume than in humans, 37.0–58.5 fl compared to 80–100 fl, respectively, the Hb concentration in both species is similar, 310–380 g l$^{-1}$. The RBCs were first sedimented by centrifugation at 2000$g$ for 5 min at 4°C. The supernatant (plasma) was a clear solution, and it and the 'buffy coat' (platelets and white blood cells) that had sedimented above the RBCs were removed by vacuum-pump aspiration. The RBC pellet was resuspended in isotonic saline–glucose (154 mM NaCl, 10 mM D-glucose, pH 7), constituted in $H_2O$ and re-centrifuged, as before. After resuspending in additional isotonic saline, the RBC suspension was bubbled with CO for 10 min, to convert the Hb–FeII into a low-spin diamagnetic state. After that it was re-centrifuged. The pellet of intact RBCs was resuspended in isotonic saline–glucose constituted in $D_2O$ (154 mM NaCl, 10 mM D-glucose, $D_2O$ 99.9%) and washed with this solution three times by centrifugation. The RBC suspension was then adjusted to haematocrit approximately 62% with the $D_2O$ saline. Monensin and valinomycin were from Sigma-Aldrich (St Louis, MO, USA) and used as supplied. Both were added to RBC suspensions in the form of concentrated dimethylsulfoxide (DMSO) solutions to a final concentration of approximately 5 µM. The amount of the DMSO added was minimized (approx. 0.5 µl in 1 ml), as at this concentration, it was found this amount of DMSO caused no substantial lysis of an RBC suspension as evidenced by haematocrit measurements.

Lysates for QENS experiments were obtained by freezing an RBC sample at 200 K in the cryostat, followed by thawing. These samples were deep red in colour but completely transparent after the experiment, which is indicative of successful lysis of the RBCs. The pH in RBCs is typically 7.4, and the pH values of the RBC samples, as well as the haemolysates, should be close to that value due to the strong buffering capacity of the Hb (12 histidyls per α and β chain with p$K_a$ values around 6.9).

## 2.2. USANS and SANS experiments

USANS was measured on the Bonse-Hart camera KOOKABURRA at ANSTO (Australian Nuclear Science and Technology Organisation) Lucas Heights, New South Wales, Australia [15]. In the experiments, a neutron wavelength of $\lambda = 4.74$ Å was used. Sample thickness was 0.5 mm, which minimized multiple scattering. Quartz sample cells were used in rotational sample tumblers that obviated RBC sedimentation. Sample temperature throughout the USANS experiments was ambient room temperature.

The raw USANS rocking curves were reduced to absolutely scaled slit smeared intensity using a data reduction protocol based on that of Kline [22], implemented in the Gumtree environment [23]. KOOKABURRA slit smeared scattering curves were then de-smeared using data reduction software [22] that relies on Lake's algorithm [24]. The resulting $I(q)$ curves, where $q$ is the scattering vector defined, $q = 4\pi \sin(\theta)/\lambda$, where $2\theta$ is the scattering angle, are directly comparable with SANS data and suitable for subsequent analysis.

USANS data of human RBCs under physiological conditions and human RBCs treated with monensin were analysed using the empirical model function referred to as 'Butler empirical' [25], specifically:

$$I(q) = I_0 \cdot \left[ \frac{\alpha}{1-\alpha} + 2\left(\frac{q\xi}{2\pi}\right)^{-d} \right] \Big/ \left[ \frac{1}{1-\alpha} + \left(\frac{q\xi}{2\pi}\right)^{-2d} \right]. \tag{2.1}$$

This function describes USANS data with one correlation peak being centred at $q^*/2\pi = 1/\xi$ with correlation length $\xi$ and a power-law range at high $q$-values with power-law coefficient $d$, while $\alpha$ is a scaling parameter.

USANS data of human RBCs treated with valinomysin that did not show a correlation peak were fitted with a generalized Guinier–Porod model as suggested by Hammouda [26], specifically

$$I(q) = I_0 \cdot G \cdot \exp\left(\frac{-q^2 R_g^2}{3}\right), \text{ for } q \leq q_1$$

and

$$I(q) = I_0 \cdot \frac{D}{q^{-d}}, \quad \text{for} \quad q > q_1. \tag{2.2}$$

where $G$ and $D$ are Guinier and Porod scaling factors, $q_1$ is calculated analytically within the model function based on $R_g$ and $d$. Details can be found in [26]. The fitted parameters of the generalized Guinier–Porod model are the radius of gyration $R_g$ and $d$ describing the power-law behaviour at high $q$-values. The generalized Guinier–Porod model as implemented in the data analysis software package jscatter was used for USANS data [27]; the Butler empirical function was implemented in Python by the authors and used as the fitting function in jscatter [27].

SANS experiments were performed on the QUOKKA [28] and BILBY [29] instruments at ANSTO. On QUOKKA, a monochromatic neutron wavelength, $\lambda$, of 5.0 Å, with a wavelength spread, $\Delta\lambda/\lambda$ of 10%, that was provided by a velocity selector was used with sample-to-detector distances of 1.3 and 10.0 m. BILBY was operated in the time-of-flight (TOF) mode with an incident polychromatic neutron beam between 2 and 20 Å. The rear detector position was 18 m, and curtain detector positions were 2 and 3 m. Protein solutions were studied in 1 mm thick quartz cuvettes. Isotropic two-dimensional scattering patterns were reduced to the form $I(q)$: using an approach based on Kline [22] in the case of QUOKKA; and in the case of BILBY, data were reduced with Mantid [30]. Sample temperature was $T = 298$ K during the experiment on QUOKKA and $T = 293$ K on BILBY. RBCs from horses were studied on QUOKKA, while human RBCs were studied on BILBY.

Hayter & Penfold derived an analytical structure factor $S(q)$ that describes the interactions between charged macro-ions, counter-ions and solvent using a rescaled mean spherical approximation (RMSA) [31,32]. An improved algorithm was developed by Belloni [33]. In this study, we used the improved algorithm as implemented in the data analysis software package jscatter [27]. For the sake of clarity in our discussions of the results, we summarize the essential equations of the RMSA algorithm.

The screened Coulomb potential between charged spherical hard-core particles in rescaled form is given by [32]

$$\beta U(x) = \gamma \exp\frac{-kx}{x}, \text{for } x > 1$$

and

$$\beta U(x) = \infty, \text{for } x < 1 \tag{2.3}$$

with the dimensionless units $x = r/\sigma$, $k = \kappa\sigma$ and $\beta = 1/k_B T$ being the inverse of thermal energy; with $k_B$ being the Boltzmann constant. The parameter values used corresponded to particle diameter $\sigma = 2R_{eff}$ and particle–particle distance $r$. We set $\kappa^{-1} = 0.304/\sqrt{I(M)} = 7.8$ Å, which is the Debye screening length in water at 293 K and 0.15 M ionic strength [34]. The term $\gamma \exp(-k)$ in equation (2.3) is the so-called contact potential in units of $k_B T$.

The effective absolute value of the surface charge $Z$ per macro-ion in units of elementary charge $e$ is calculated from the reduced parameters as

$$Z = (2+k)\sqrt{\frac{\pi\varepsilon\varepsilon_0\sigma\gamma\exp(-k)}{\beta e^2}}, \tag{2.4}$$

where $\varepsilon_0$ is the vacuum electrical permittivity and $\varepsilon$ the dielectric constant of the solvent [31,32].

Another important parameter that is contained in the RMSA algorithm is the particle volume fraction $\rho$. Finally, the intracellular Hb concentration $c$ is determined by $c = \rho / \upsilon$, where the partial specific volume of Hb is $\upsilon = 0.75$ g [5].

## 2.3. QENS experiments

QENS data were recorded on the cold neutron high-resolution backscattering spectrometer, EMU, at ANSTO [35]. This uses a final neutron wavelength of 6.27 Å and achieves a full-width half maximum energy resolution of 1.0 µeV over a $q$-range from 0.34 to 1.92 Å$^{-1}$. In our experiments, we used the total accessible energy transfer range of ±31 µeV for experiments on horse RBCs, with a binning step of 0.2 µeV. For human RBCs, we used a restricted energy transfer range of ±20 µeV with a binning step of 0.3 µeV, to improve the statistical power of the recorded data. RBC samples and buffers were measured in annular aluminium sample holders with a gap thickness of 0.2 mm; a 0.85 ml sample volume was used for all measurements. RBC samples, buffers, vanadium reference sample and empty aluminium sample holders were measured from 18 to 24 h each. The contribution of the $D_2O$ buffer to the QENS spectra of the RBC samples was subtracted. The calculated neutron transmission of a 33% Hb solution in $D_2O$ with 0.4 mm total path length is 93% for 6.27 Å neutrons.

For analysis of the QENS data, a model-independent approach was used, which is considered standard for the analysis of the dynamics of proteins in solutions [7,20,36]. The dynamical structure factor for the internal dynamics of proteins is expressed as:

$$S_I(q, \omega) = A_0(q)\, \delta(\omega) + [1 - A_0(q)]\, L_I(q, \omega), \tag{2.5}$$

where $A_0(q)$ is the elastic incoherent structure factor (EISF) that describes localized and confined motions in the protein, and $L_I(q, \omega)$ is one effective Lorentzian according to

$$L_I(q, \omega) = \frac{1}{\pi} \frac{\Gamma(q)}{\Gamma^2(q) + (\hbar\omega)^2} \tag{2.6}$$

which describes the quasi-elastic broadening that is caused by diffusive motion in the macromolecule.

Proteins in solution undergo both internal motion and global rotational and translational diffusion. The total scattering function $S_{\text{total}}(q, \omega)$ can be viewed as a convolution of two scattering functions $S_I(q, \omega)$ and $S_G(q, \omega)$ that account for global diffusion and internal dynamics of the protein. The convoluted dynamical structure factor is then given by

$$S_{\text{total}} = A_0(q)\, L_G(q, \omega) + [1 - A_0(q)]\, L_{G+I}(q, \omega). \tag{2.7}$$

Here, $L_G$ and $L_{G+I}$ are two Lorentzians with

$$L_G(q, \omega) = \frac{1}{\pi} \frac{\Gamma_G(q)}{\Gamma_G^2(q) + (\hbar\,\omega)^2} \tag{2.8}$$

and

$$L_{G+I}(q, \omega) = \frac{1}{\pi} \frac{\Gamma_G(q) + \Gamma_I(q)}{(\Gamma_G(q) + \Gamma_I(q))^2 + (\hbar\omega)^2}. \tag{2.9}$$

The line-width $\Gamma_G(q)$ of the narrow Lorentzian $L_G(q, \omega)$ contains quantitative information on both global translational and rotational diffusion. The broad Lorentzian $L_{G+I}(q, \omega)$ represents the contribution of both internal diffusive motion and global protein diffusion. The term $\Gamma_I(q)$ accounts for internal diffusive processes in the protein and its value is extracted from the broad Lorentzian by subtraction of the narrow $\Gamma_G(q)$ component. During QENS data analysis, the theoretical model function given in equation (2.7) was numerically convolved with the instrumental resolution function measured on a vanadium sample and fitted to the QENS spectra, including a linear background for the noise and fast dynamics that could not be resolved by the instrument.

## 3. Results and discussion

Hb–Hb interactions, as well as global diffusion of Hb in RBCs, should change in response to changes in cell volume, intracellular Hb concentration and related morphological variations in the cell. First, RBC shape/volume changes and the resulting modification of Hb–Hb interactions were investigated using

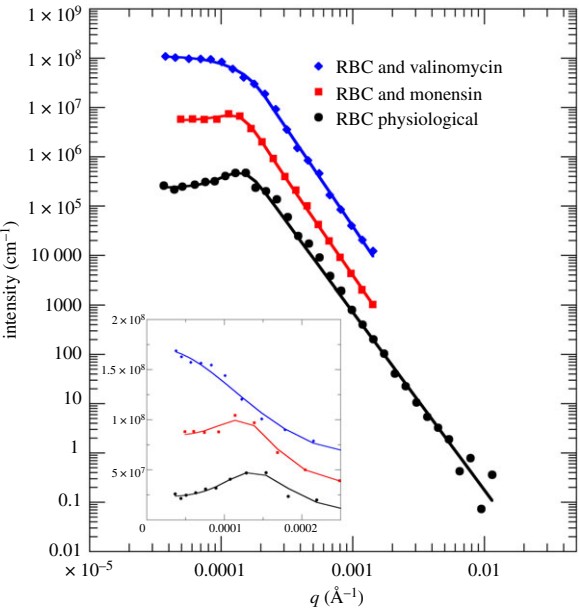

**Figure 1.** De-smeared USANS data of human RBCs under physiological conditions and treated with valinomycin or monensin. A focus at the small $q$-range is shown as inset to highlight the absence of the correlation peak for human RBC with added valinomycin. Symbols denote experimental data; solid lines are regression fits to the data. The model function 'Butler empirical' was used to fit the USANS data of human RBCs and human RBCs with monensin (see text); USANS data of human RBCs and valinomycin were fitted with a generalized Guinier–Porod model (see text). USANS data were measured on KOOKABURRA at ANSTO.

USANS/SANS experiments on the ANSTO instruments KOOKABURRA, BILBY and QUOKKA. The corresponding modification of Hb dynamics was investigated using QENS on EMU on the angstrom and nanosecond length and time scales.

$Na^+$ and $K^+$ concentrations in the RBCs are regulated by Na,K-ATPase that hydrolyses ATP in the process of maintaining $Na^+$ and $K^+$ concentration gradients across the cell membrane [8]. The cytoplasmic concentration of $K^+$ in RBCs is substantially higher than in the blood plasma or isotonic saline (NaCl) medium [37]. This situation is the opposite for $Na^+$ ions, having a lower concentration in the RBC interior and a higher concentration in the blood plasma, or in our cases, the isotonic saline solution used here [38]. In our work, RBC morphology was altered by perturbing the metabolic disequilibrium of $K^+$ in the cytoplasm, and $Na^+$ in the extracellular medium. For this purpose, the two ion-selective antibiotic ionophores monensin and valinomycin were added to RBC samples.

Monensin and valinomycin each bind one monovalent cation and transport it across the cell membrane [39]. Monensin binds preferentially to $Na^+$ [40], while valinomycin binding occurs preferentially to $K^+$ [41]. As a consequence, the $Na^+/K^+$ balance between the RBC cytoplasm and plasma is altered, resulting in modified osmotic pressure in the cells. This leads to RBC expansion or shrinkage depending on the experimental conditions: monensin causes RBC expansion when they are in an NaCl–saline solution due to increased cellular $Na^+$ concentration; while valinomycin results in RBC shrinkage due to lowering of cytoplasmic $K^+$ concentration when the cells are suspended in NaCl–saline. Neutron scattering from RBCs in isotonic NaCl–$D_2O$ saline was first measured, thus providing a reference set of data.

## 3.1. RBC morphology and Hb–Hb interactions investigated by USANS/SANS

Human RBC suspensions were studied on the USANS instrument KOOKABURRA to investigate the effect of both ionophores on structural properties of RBCs on the micrometre length scale. De-smeared USANS data are shown in figure 1. While all RBC samples exhibited a power-law scattering behaviour at large $q$-vectors with power-law coefficient $d$, only RBC in isotonic saline, and RBC with added monensin show an additional correlation peak in the low $q$-range at $q^* = 2\pi/\xi$. In these samples, there was a well-defined correlation length $\xi$ (see inset of figure 1 for a zoom-in). This regular arrangement was not seen with USANS data of RBCs that were treated with valinomycin.

The USANS data from RBCs in isotonic saline and RBCs treated with monensin were fitted with the model function Butler empirical (equation (2.1)) that allowed the estimation of the correlation lengths $\xi$ and power-law coefficients $d$. We estimated that for RBCs in their physiological state $\xi = 4.61 \pm 0.10$ μm and $d = 3.6 \pm 0.1$, while for RBCs treated with monensin, we found $\xi = 5.28 \pm 0.08$ μm and $d = 3.8 \pm 0.1$. USANS data of RBCs treated with valinomycin were fitted using the generalized Guinier–Porod model (equation (2.2)) that yields information on the Guinier radius $R_g$ and power-law coefficient $d$. We estimated that $R_g = 1.15 \pm 0.06$ μm and $d = 3.9 \pm 0.1$ for RBCs with valinomycin. However, we note that structure factor effects in the densely packed RBC suspensions were evident; USANS experiment with more dilute RBC suspensions would be needed to clarify this feature. However, we could clearly see the absence of a correlation peak $q^*$ in the RBCs that were treated with valinomycin.

How can we explain these USANS findings? USANS is most sensitive to the shape and packing interactions of whole RBCs. Contributions arising from Hb–Hb interactions (structure factor) and Hb form factor scattering can be neglected, as they appear as a quasi-flat background in the USANS distance range. Due to their biconcave disc shape (discocyte) RBCs, in static conditions, arrange in ordered stacks called rouleaux [42]. Therefore, we attribute the observed correlation peaks $q^*$ to rouleaux formation in the RBC suspensions at high haematocrit. The correlation length increases from $\xi = 4.61$ μm (RBC physiological shape) to $\xi = 5.28$ μm (RBC with monensin) due to the swelling of the cells, albeit the discocytic shape appears to remain intact during the expansion still causing their ordering. The correlation peak disappears entirely for RBCs treated with valinomycin; and we obtained an effective Guinier radius of $R_g = 1.15$ μm of the shrunken RBCs. That $R_g$ would correspond to an effective radius $R$ of the RBC assuming the spherical shape of the shrunken RBC with $R = \sqrt{(5/3)} \cdot R_g \approx 1.5$ μm.

The volume of such a shrunken spherical RBC achieved with valinomycin would be approximately $V = 4\pi R^3/3 = 14$ fl. The average volume of a human RBC under physiological conditions is $V = 88$ fl [43]. Therefore, our USANS results demonstrate the morphological shape change from discocytes to echinocytes or spherocytes; and the shrunken size of RBCs upon addition of valinomycin. However, we needed to be careful with further quantitative statements concerning the RBC volume determination by USANS as interparticle effects were present at high haematocrit values, which modified the obtained $R_g$ value. Hence, by considering the intracellular Hb concentration obtained by SANS and QENS (see below for details), the cell volume of RBCs treated with valinomycin was underestimated by a factor of approximately 5. Furthermore, USANS studies using diluted RBC suspensions would be needed to clarify this aspect of the analysis. The power-law coefficients of all RBC samples ($d = 3.6$–3.9) were close to the Porod limit of $d = 4$, and we interpreted the power-law scattering of the USANS data to arise from the smooth RBC surface [44].

As a next step, we have performed additional SANS experiments on the instruments QUOKKA and BILBY at ANSTO to obtain information at the nanometre length scale and to investigate the effect of RBC shape changes on Hb–Hb interactions. RBC from horse has been investigated on QUOKKA, while human RBC has been studied on BILBY. A monochromatic neutron beam was used on QUOKKA and two detector positions were needed to cover the full $q$-range, while BILBY was operated in the TOF mode using a polychromatic incident neutron beam and the full experimental $q$-range could be covered in a single instrument configuration. Hence, we could use the TOF SANS mode on BILBY to perform kinetic SANS experiments as a function of incubation time and to investigate the stability of the RBC samples over an extended time range. Experimental SANS data of human RBC and horse RBC under physiological conditions and incubated with the ionophores monensin and valinomycin directly after mixing are shown in figure 2.

QUOKKA and BILBY SANS data were fitted using the function

$$I(q) = I_1 \cdot q^{-d} + I_2 \cdot F(q) \cdot S(q). \tag{3.1}$$

The first term in equation (3.1) is needed as there is some forward scattering that could be described effectively by a power-law function with power-law coefficient $d$. The second term in equation (3.1) describes the contribution of the Hb form factor $F(q)$ that is modulated by the structure factor $S(q)$ arising from Hb–Hb interactions. The theoretical form factor $F(q)$ was calculated from the crystal structures of deoxy horse Hb [45] (pdb ID: 2DHB) and deoxy human Hb [46] (pdb ID: 2DN2) using the computer program CRYSON [47]. It was shown by Krueger & Nossal [2] that Hb–Hb interactions in concentrated Hb solutions can be described effectively by a screened Coulomb potential between charged spherical hard-core particles. Hence, we used the RMSA approach developed by Hayter & Penfold [31,32] to calculate the corresponding structure factor $S(q)$ for that interaction potential (see

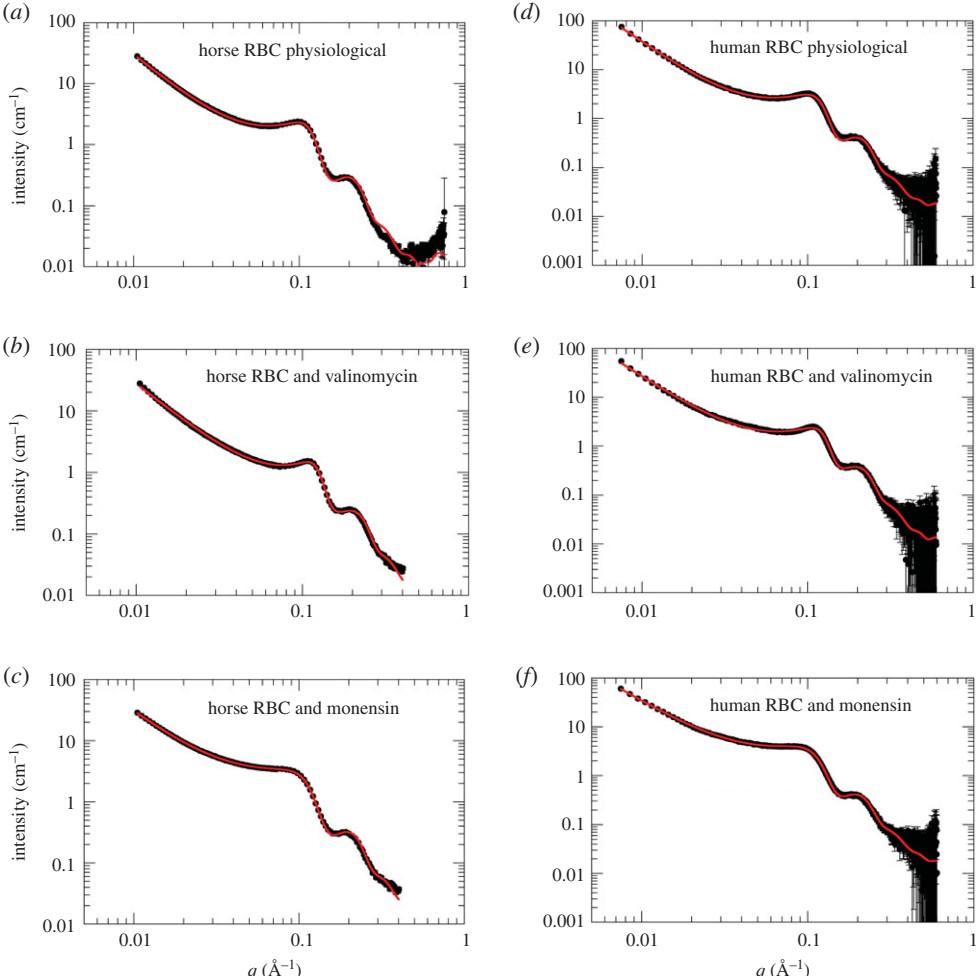

**Figure 2.** Experimental SANS data of the investigated RBC samples. ($a$–$c$) Horse RBC samples measured on the instrument QUOKKA. ($d$–$f$) Human RBCs investigated on BILBY. Symbols denote experimental data; solid red lines are theoretical fits using equation (3.1). The broadening of the first interaction peak at approximately 0.1 $\text{Å}^{-1}$ upon addition of monensin ($c,f$) is a clear sign of the lower intracellular Hb concentration in the swollen RBCs, while the narrowing of that first interaction peak ($b,e$) was caused by the increase in the intracellular Hb concentration in the valinomycin-treated RBCs.

Material and methods, and USANS and SANS experiments for details). Fits to the SANS data using that theoretical model are shown in figure 2.

The obtained power-law coefficients are $d = 2.22 \pm 0.01$ for horse RBC under physiological conditions, $d = 2.03 \pm 0.02$ for horse RBC and monensin and $d = 2.16 \pm 0.01$ for horse RBC and valinomycin. Power law coefficients for human RBC in their physiological state are $d = 2.35 \pm 0.01$, $d = 2.01 \pm 0.01$ for human RBC and monensin, and $d = 2.21 \pm 0.01$ for human RBC and valinomycin. Power-law coefficients with $d < 3$ are characteristic signs for mass fractal morphologies [48], while scattering from flat lipid membranes would yield a power-law coefficient of $d = 2$. However, due to the extremely high relative concentration of Hb in RBC, the scattering contribution of the RBC membrane gives only a relatively weak signal compared to the Hb signal due to its extremely low volume fraction in the solution. In a previous SANS study on concentrated Hb solutions, a power-law coefficient of $d = 2.67 \pm 0.01$ has been reported for a 290 mg ml$^{-1}$ Hb solution [49] and it was attributed to the presence of a large superstructure with characteristic size extending to several hundred nanometres.

The power-law scattering contribution was subtracted from the experimental data shown in figure 2 and in a second step, the data were divided by the theoretical form factor $F(q)$ based on crystallographic data. The shape of the structure factors $S(q)$ is then clearly revealed and the $S(q)$ obtained using that approach are plotted in figure 3 together with fits using the theoretical model treating the Hb–Hb interaction potential as charged hard-spheres with screened Coulomb interaction. The theoretical model offers a good description of the experimental $S(q)$ up to the first peak maximum, then it appears to deviate slightly for larger $q$-values. An explanation might be that Hb–Hb interactions are

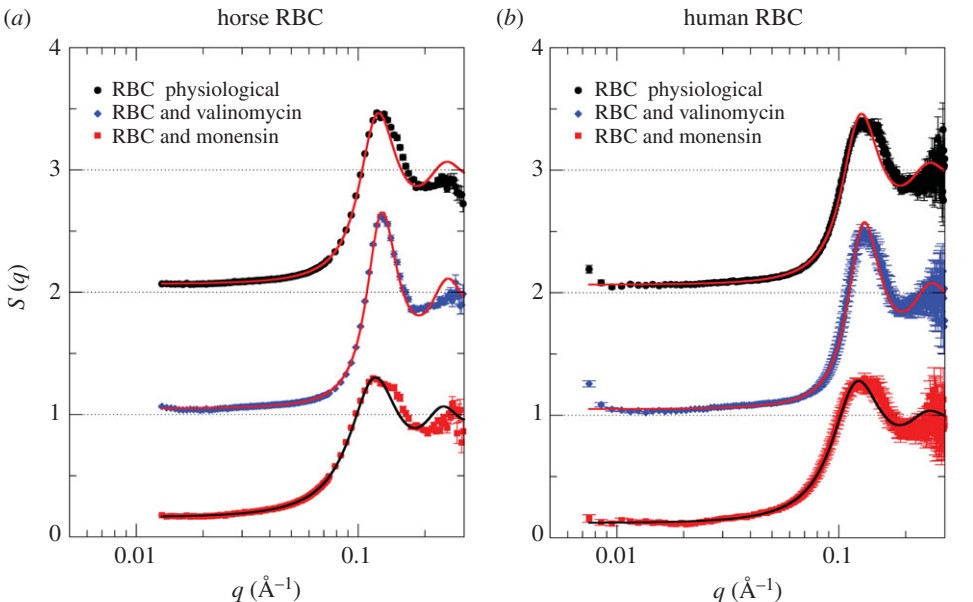

**Figure 3.** Structure factors describing Hb–Hb interactions in the investigated RBC samples. (*a*) Horse RBCs investigated with QUOKKA; (*b*) human RBCs studied with BILBY. Experimental structure factors $S(q)$ were calculated from SANS data shown in figure 2 (see text for details). Solid lines are fits to the data treating Hb as charged hard-spheres with screened Coulomb interactions.

not perfectly modelled as screened hard-spheres and, for example, attractive interactions would need to be included for a perfect description of the $S(q)$. We refer here to future studies to clarify that aspect regarding detailed theoretical modelling of Hb–Hb interactions. Note that the information content in figures 2 and 3 is identical.

All physical parameters obtained from the structure factor fits of horse and human RBC in dependence of RBC shape are compiled in table 1. In addition, we have added the parameters reported by Krueger & Nossal [2] to table 1 describing the Hb–Hb interactions in human RBC as charged hard-spheres as a function of buffer osmolarity.

In agreement with the original study by Krueger & Nossal [2] on Hb–Hb interactions in RBC, we find that under physiological conditions and in the shrunken RBC induced by addition of valinomycin, the contact potential has a value of a few $k_BT$ corresponding to an effective surface charge of 15–18 elementary charges per Hb. A perturbation of the Hb–Hb interaction potential is observed in the expanded RBC treated with monensin. Here, the contact potential and, hence, the effective surface charge per Hb is significantly reduced, which was observed by Krueger & Nossal as well for expanded RBC in 150 mOsM buffer [2]. At reduced Hb concentration, the Hb–Hb interactions become closer to ideal hard-sphere behaviour. We attribute this observation to the larger average distance between the Hb molecules at lower concentration and the larger intracellular $Na^+$ concentration, both of which could effectively restrict the electrostatic repulsion due to charge screening. The obtained effective hard sphere radii of Hb are slightly smaller than reported by Krueger & Nossal, but still in a reasonable and uniform range. Additionally, we obtained the Hb volume fraction $\rho$ and intracellular Hb concentration $c$ from the structure factor fits. The intracellular Hb concentration is a very valuable parameter and provides insights into RBC shape changes. Concerning human and horse RBC, we obtain intracellular Hb concentrations between 330 and 340 mg ml$^{-1}$, which are close to the expected value [6]. Concerning the effect of the ionophores on RBC morphology, we can now directly see that valinomycin treatment increases the intracellular Hb concentration due to RBC shrinkage, while incubation with monensin causes a reduction of intracellular Hb concentration corresponding to RBC expansion. The Hb concentration changes can be translated into RBC volume changes and treatment of RBC with both ionophores results in RBC volume changes of around 20% when considering the average values of both human and horse RBC.

In a further step, we investigated the stability of the RBC in the different morphological shapes. As we have seen above, intracellular Hb concentration is informative on RBC volume. Therefore, we performed kinetic TOF SANS experiments on BILBY to determine intracellular Hb concentration in human RBC as a function of incubation time. We analysed the experimental TOF SANS data in the identical way as described above. The extracted structure factors at selected incubation time points are shown in figure 4.

**Table 1.** Parameters describing Hb–Hb interactions in horse and human RBC as charged hard-spheres determined from SANS experiments.

| | contact potential ($k_B T$) | $Z$ ($e$) | $R_{eff}$ (Å) | $\rho$ | $c$ (mg ml$^{-1}$) |
|---|---|---|---|---|---|
| horse RBCs[a] (physiological) | 2.5 ± 0.4 | 16 ± 1 | 23.6 ± 0.5 | 0.256 ± 0.015 | 341 ± 20 |
| horse RBCs + valinomycin[a] (shrunken cells) | 1.9 ± 0.3 | 15 ± 1 | 24.1 ± 0.3 | 0.319 ± 0.009 | 426 ± 12 |
| horse RBCs + monensin[a] (expanded cells) | 0.1 ± 0.1 | 4 ± 4 | 25.6 ± 0.2 | 0.229 ± 0.003 | 305 ± 5 |
| human RBCs[a] (physiological) | 3.3 ± 0.1 | 18 ± 1 | 22.6[c] | 0.249 ± 0.001 | 332 ± 2 |
| human RBCs + valinomycin[a] (shrunken cells) | 3.1 ± 0.1 | 17 ± 1 | 22.6[c] | 0.279 ± 0.001 | 372 ± 1 |
| human RBCs + monensin[a] (expanded cells) | 1.3 ± 0.1 | 12 ± 1 | 22.6 ± 0.6 | 0.224 ± 0.001 | 253 ± 19 |
| human RBCs normal salt[b] (300 mOsM, physiological) | 3.6 | 12 | 27.8 | 0.223 | 297 |
| human RBCs high salt[b] (600 mOsM, shrunken cells) | 5.6 | 20 | 26.8 | 0.286 | 381 |
| human RBCs low salt[b] (150 mOsM, expanded cells) | 0.8 | 4 | 28.4 | 0.166 | 221 |

[a]This study: horse RBCs used QUOKKA; human RBC samples were studied on BILBY.

[b]Data from Krueger & Nossal [2]. Contact potential values reported by Krueger & Nossal scaled by a factor of 4 to be directly comparable with the present study.

[c]Parameter $R_{eff}$ obtained from fits of human RBCs + monensin were used for human RBCs and human RBCs + valinomycin, and kept fixed during fits.

Concerning the expanded RBC, with added monensin, we found perturbations of the $S(q)$ that were first visible after 13 h incubation time (figure 4c), which further deteriorated after 24 h (data not shown). This was ascribed to RBC lysis, which produced two compartments for Hb: a dilute extracellular compartment; and a more concentrated solution confined in the RBCs. Hence, we restricted further QENS investigations (see below) to a maximum total measurement time of 24 h.

As a probe of RBC shape stability, we estimated the intracellular Hb concentrations (figure 5) for RBCs with physiological shape and both shrunken and expanded forms that were induced by addition of valinomycin and monensin, respectively. A TOF SANS measurement on BILBY required 30 min per sample. We noted that within this time window, all RBC shape changes had already occurred within the first recorded time point. Furthermore, we found a stability of the monensin-treated RBC sample over an incubation time of 1 day, while the RBC under physiological conditions and the RBC with valinomycin remained stable for at least 33 h in the neutron beam at 293 K; this was the longest recorded incubation time. Average values of Hb concentration are reported in figure 5; they provided us with reliable and accurate reference data on intracellular Hb concentrations.

Finally, the kinetic TOF SANS data validated the RBC sample stability over the long beam time that was needed for the QENS experiments. We used these samples to investigate Hb diffusion and dynamics in RBCs that had altered morphology. In the following section, we present our experimental QENS data and discuss the observed Hb diffusion coefficient in the context of intracellular Hb concentration and Hb–Hb interactions as estimated from the SANS data.

## 3.2. Hb dynamics probed by QENS experiments

We investigated the effect of adding the ionophores on the diffusion coefficient of Hb using this parameter as a reporter of intracellular Hb concentration (and hence an indicator of Hb–Hb interaction). The diffusion coefficient of Hb in an RBC lysate was also measured. The experimental QENS data from horse and human RBC samples in isotonic D$_2$O buffer are shown in figure 6 over an energy transfer range of ±20 µeV at a scattering vector of $q = 0.91$ Å$^{-1}$. Horse RBCs were studied at $T = 288$ K, while human RBCs were at $T = 298$ K. This variation of temperature verified the absence (or minimal dependence) of the measurements on the 10 K temperature difference. The RBC QENS data were analysed according to equation (2.7) by a superimposed narrow and a broad Lorentzian. Thus,

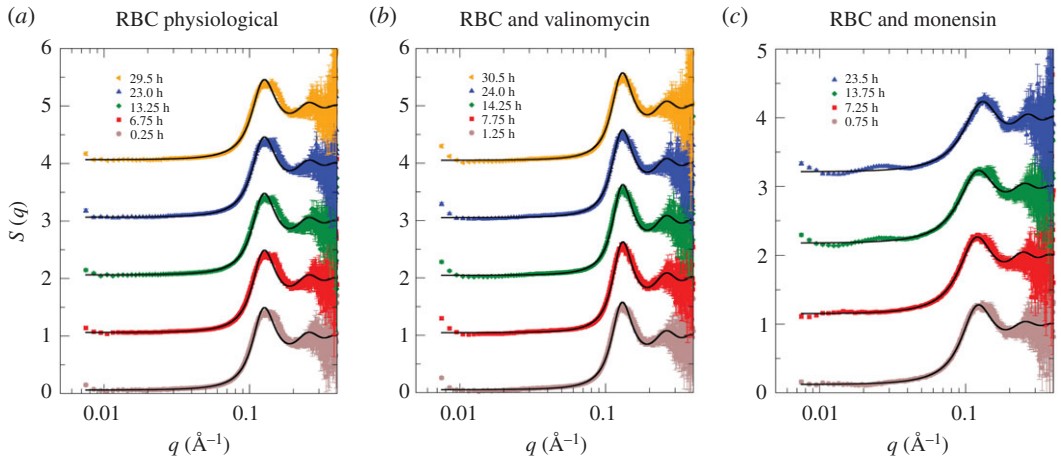

**Figure 4.** Structure factors of (*a*) human RBCs under physiological conditions and (*b,c*) with added ionophores valinomycin and monensin. Structure factors were estimated from TOF SANS data measured on BILBY as a function of the incubation time. Symbols denote experimental data at selected incubation time points; solid lines are theoretical fits, treating the Hb–Hb interaction as charged hard-spheres with screened Coulomb interactions.

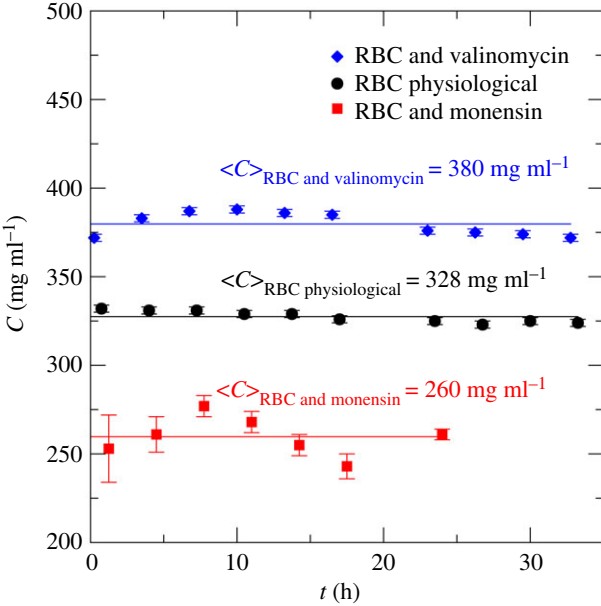

**Figure 5.** Intracellular Hb concentration in human RBCs in their physiological state and human RBCs treated with the ionophores valinomycin and monensin as a function of incubation time *t*. Hb concentration was estimated from theoretical fits of structure factors determined by using TOF SANS experiments on BILBY. Symbols denote experimentally determined intracellular Hb concentration; solid lines indicate the intracellular Hb concentrations $\langle C \rangle$ averaged over the explored incubation time window.

information on both global Hb diffusion coefficient and internal Hb dynamics were derived from the line-widths at half-maximum height (HWHM) $\Gamma_G(q)$ and $\Gamma_I(q)$, respectively.

Measured line-widths $\Gamma_G(q)$ of representative horse and human RBC samples under native conditions, and with the two added ionophores as well as the RBC lysate are given in figure 7. These data are shown as a function of the squared scattering vector, $q^2$. Apparent diffusion coefficients were estimated from the line that was fit to the data using the expression $\Gamma_G(q) = D_{eff} q^2$. The $D_{eff}$ estimated by QENS contained contributions from both rotational and translational diffusion. For spherical-shaped particles, such as Hb, where rotational and translational diffusion are not correlated, the $D_{eff}$ estimated from QENS data are 1.27 times larger than pure centre-of-mass self-diffusion [20,50]. The $D_{eff}$ of Hb estimated from QENS experiments are given in table 2 for horse RBC at $T = 288$ K and human RBC at $T = 298$ K. Further below, we used $D_{eff}$ values corrected for the contribution of

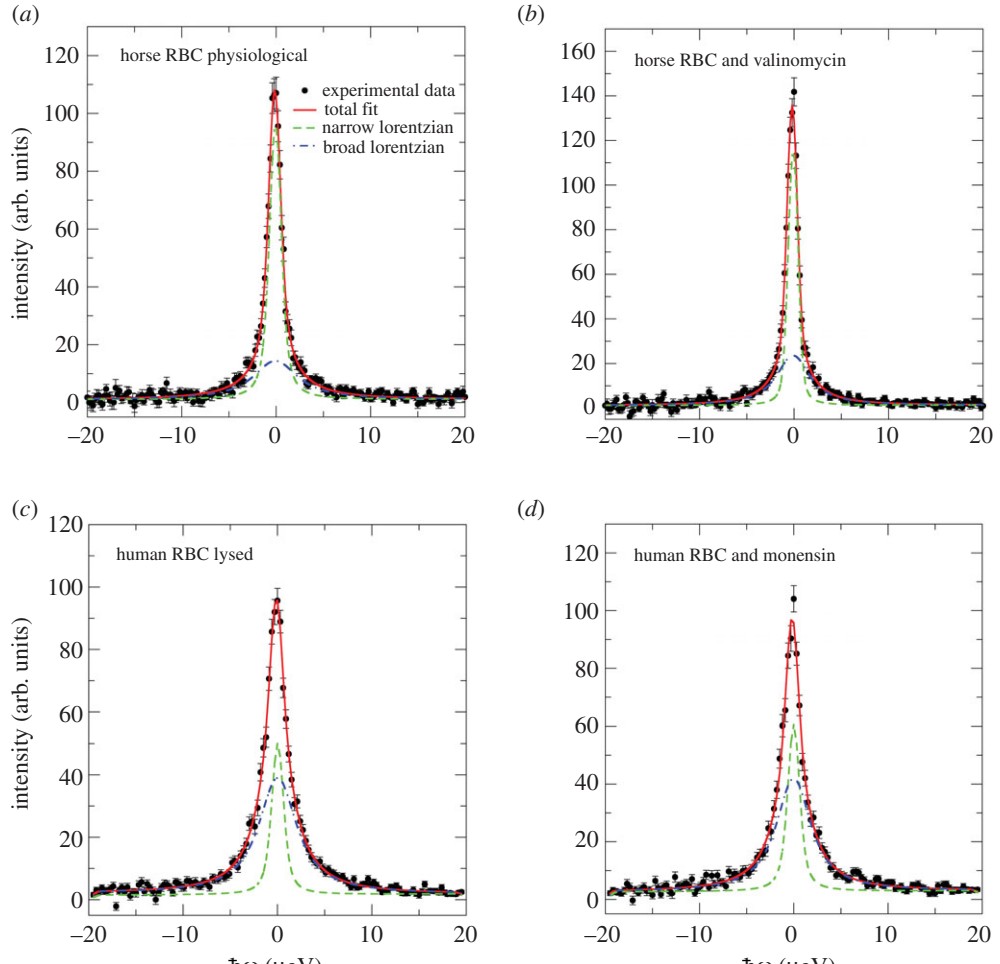

**Figure 6.** Representative QENS spectra of (*a*) horse RBCs, (*b*) horse RBCs treated with valinomycin, (*c*) human RBC lysate and (*d*) human RBCs treated with monensin. QENS data were measured with the EMU neutron backscattering spectrometer. Temperature was $T = 288$ K for horse RBCs and $T = 298$ K for human RBCs. All QENS spectra are shown for $q = 0.91$ Å$^{-1}$. Solid symbols were experimental data, the red lines denote the total fit function, while the narrow and broad Lorentzians are shown as green dashed and blue dashed-dotted lines, respectively. All functions were convolved with the instrument's resolution function.

rotational diffusion by $D_{eff}/1.27$ before insertion in equation (3.5) (see §3.3). We inferred a strong effect of valinomycin and monensin on the overall (global) apparent diffusion coefficient of Hb in both types of RBC. Valinomycin resulted in shrinkage of the cells and, hence, elevated intracellular Hb concentration. A reduction in the apparent Hb diffusion coefficient was consistent with this change. Monensin, on the other hand, has the opposite effect on RBC volume, so it leads to swelling of the cells; consequently, intracellular Hb concentration is reduced. This outcome was directly inferred from the increase in the measured Hb diffusion coefficient, compared with the reference sample measured under physiological buffer conditions.

To perturb Na,K-ATPase in the RBC membranes, we increased the temperature of the previously measured monensin-treated RBC to $T = 308$ K and incubated the cell suspension for 5 h. Thus, the RBCs metabolized the glucose in the medium more quickly. The temperature was then lowered to 298 K and a new QENS measurement was begun on the same RBCs, with monensin being present. Within the experimental uncertainty, no differences in the measured global diffusion coefficient of Hb were seen between fresh monensin-treated RBCs, and swollen RBCs that had consumed the available glucose. This finding validated the notion that monensin alone was able to induce the maximal shape change (swelling) of the RBCs. After lysing the RBCs by freeze–thawing, however, a significantly larger diffusion coefficient was recorded. This demonstrated that monensin-treated RBCs were in their fully swollen state and further expansion would have resulted in cell lysis.

Representative line widths $\Gamma_I(q)$, which provide information on the internal diffusive motion in Hb of horse and human RBCs under different solvent conditions, are given in figure 8. $\Gamma_I(q)$ shows

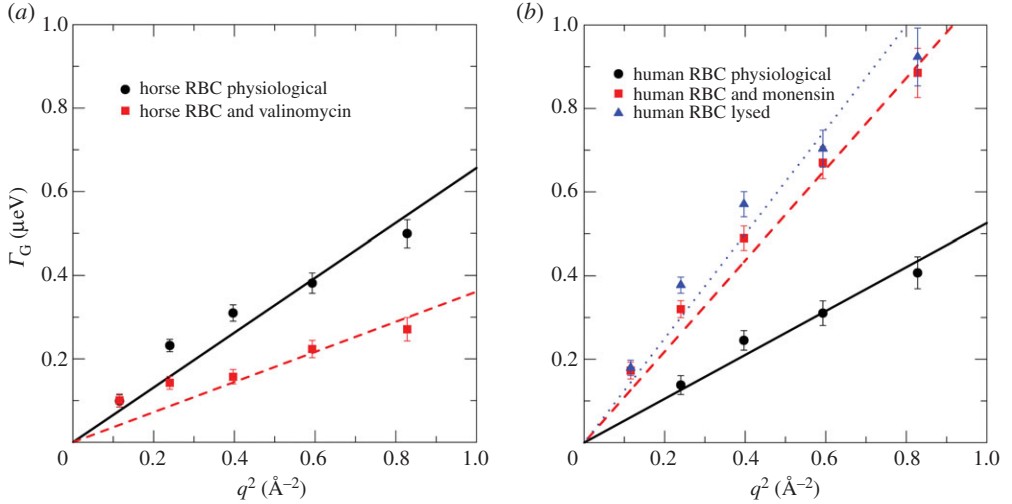

**Figure 7.** Line-widths $\Gamma_G$ as a function of $q^2$ describing the global Hb diffusion coefficient in RBCs and in a haemolysate. (a) Data from horse RBCs at $T = 288$ K. (b) Data from human RBCs at $T = 298$ K. Symbols are experimental data and straight lines are linear fits to the data. Apparent Hb diffusion coefficients $D_{eff}$ were estimated from the slopes of the lines. Errors were obtained from least square fits to the QENS data.

**Table 2.** Global diffusion and internal dynamics of Hb from horse and human.

|  | $D_{eff}$ ($\times 10^{-8}$ cm$^2$ s$^{-1}$) | $\langle \Gamma \rangle$ ($\mu$eV) | $\tau$ (ps) |
|---|---|---|---|
| horse RBCs at $T = 288$ K |  |  |  |
| RBCs physiological state | $10.0 \pm 0.3$ | $2.84 \pm 0.96$ | $232 \pm 78$ |
| RBCs + valinomycin | $5.5 \pm 0.3$ | $2.34 \pm 0.92$ | $281 \pm 111$ |
| human RBCs at $T = 298$ K |  |  |  |
| RBCs physiological state | $8.0 \pm 0.4$ | $2.40 \pm 1.32$ | $274 \pm 150$ |
| RBCs + monensin | $16.6 \pm 0.5$ | $3.68 \pm 1.80$ | $179 \pm 88$ |
| RBCs + monensin, glucose reduced metabolically | $16.2 \pm 0.4$ | $3.50 \pm 1.16$ | $188 \pm 62$ |
| RBC lysate | $19.0 \pm 0.5$ | $2.48 \pm 0.56$ | $265 \pm 60$ |

$q^2$-independent behaviour within experimental error, and there is no proportionality with $q^2$. Thus, internal restricted and localized motions were seen. Average values $\langle \Gamma \rangle$ were calculated from the measured $\Gamma_I(q)$ data, and correlation times that informed us on internal Hb dynamics. These were calculated by using $\tau = \hbar \langle \Gamma \rangle$; the values of $\langle \Gamma \rangle$ and $\tau$ are given for the measured samples of horse RBCs at $T = 288$ K and human RBCs at $T = 298$ K in table 2.

No significant impact of Hb shape changes on internal Hb correlation times was observed within the accuracy of our experiments. Thus, we concluded that global protein diffusion and internal Hb dynamics were uncorrelated for the protein-solution samples. The estimated average value for the correlation time of horse Hb was $\langle \tau \rangle = 246 \pm 31$ ps, from native RBCs and valinomycin-treated RBCs at 288 K; an average value of $\langle \tau \rangle = 227 \pm 50$ ps was estimated for all human RBC samples at 298 K. As expected, these correlation times were not strongly dependent on temperature, as has been reported before [51].

We can compare our results to other experimental studies that were performed using high-resolution neutron backscattering spectroscopy at other international neutron sources. Correlation times of Hb reported here are in quantitative agreement with the values found for $\alpha$-helical myoglobin in different folded states [52]. And they are of the same order of magnitude, but systematically larger, than those in a mixed $\alpha/\beta$ photosensor protein [52], and in streptavidin that consists predominately of $\beta$-sheet barrels [53]. As QENS probes average dynamics within proteins, we cannot conclude from QENS in an easy way, which residues contributed to the measured signal, and to what extent. Here, residue-resolved nuclear magnetic resonance (NMR) experiments should provide the required information.

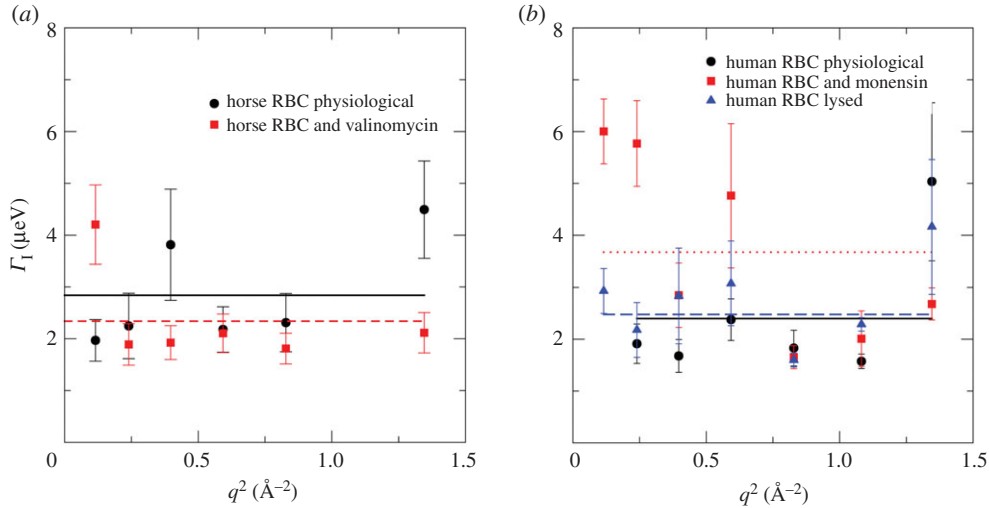

**Figure 8.** Line-widths $\Gamma_1$ as a function of $q^2$ accounting for Hb internal diffusion in: (a) horse RBCs at $T = 288$ K and (b) human RBCs at $T = 298$ K.

Motions probed by high-resolution QENS on the nanosecond time scale have been associated with motions of the protein backbone, and of slow rotational and confined motions of amino acid side chains. As the measured correlation times report on the rate of thermal conformational fluctuations, they are an indication of the energy landscape in the protein that is generated by its folded structure. Larger correlation times are a characteristic of slower motions. Hence, motions on the nanosecond time scale, as seen by high-resolution QENS, are slower in α-helical proteins, and become progressively faster in proteins that contain β-sheets. This behaviour could be due to the large number of hydrogen bonds that are formed between side chains in a α-helical structure; however, such a hypothesis requires further rigorous study. Concerning dynamics on the picosecond time scale, a variation of secondary structure mostly affects the EISF, while the correlation times of internal motions remain unaffected [54,55].

Information on the extent of average motions in Hb could be obtained from the EISF as defined in equation (3.2). Due to counting statistics, we could determine the EISF only for the horse RBCs at 288 K. The EISF was interpreted using the model function for localized motions in confinement with Gaussian statistics [56]

$$\text{EISF} = A_0(1 - p) \cdot \exp(-\langle x^2 \rangle \cdot q^2) + p, \tag{3.2}$$

where $\langle x^2 \rangle$ is the mean square displacement of Hb, and $p$ is the fraction of hydrogen atoms that move too slowly and appear as immobile within the time resolution of the EMU spectrometer. The measured EISF and the fits with the theoretical model are shown in figure 9. For RBCs in their physiological state, we obtained $\langle x^2 \rangle = 1.3 \pm 1.0$ Å$^2$ and $p = 0.50 \pm 0.13$ and for RBCs + valinomycin $\langle x^2 \rangle = 0.9 \pm 0.8$ Å$^2$ and $p = 0.35 \pm 0.27$ at 288 K. We previously reported amplitudes of motion of Hb in RBCs of $r = 2.8 \pm 0.1$ Å at 290 K, by using a diffusion in a sphere model for the EISF with sphere radius $r$ [7]. As mentioned by Volino et al., the conversion between the diffusion in a sphere and the Gaussian model for the EISF gives $\langle x^2 \rangle = r^2/5$ [56]. Hence, the previously published spherical radii, $r$, for the extent of internal motions of Hb in RBCs corresponded to $\langle x^2 \rangle = 1.6 \pm 0.1$ Å$^2$ at 290 K, which is compatible with the values reported in the present study.

In the following section, we focus on the effect of RBC shape changes on apparent (global) Hb diffusion in RBCs and make a direct link back to the SANS results informing on Hb–Hb interactions.

## 3.3. Interpretation of QENS data on global Hb diffusion

Longeville & Stingaciu [6] note that short-time Hb diffusion in RBCs can be described by a simple equation that is derived for hard-sphere particles [57] according to

$$\eta(c) = \eta_0 \exp\left(\frac{[\eta]c}{1 - k/v[\eta]c}\right) \tag{3.3}$$

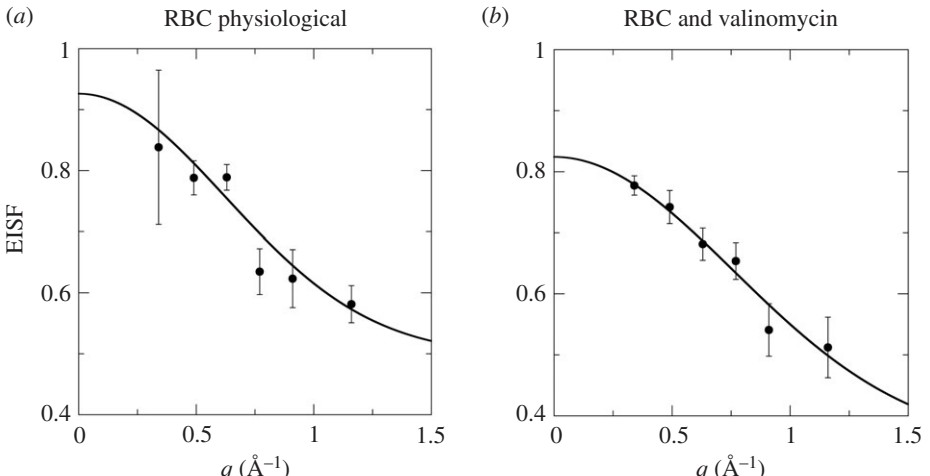

**Figure 9.** EISF of confined motions of Hb in horse RBCs at 288 K. (*a*) RBC physiological state and (*b*) RBCs + valinomycin. Symbols denote experimental data; lines are fits according to equation (3.2) to extract the mean square displacements of average Hb motions; and error bars denote 1 s.d.

where $\eta(c)$ is the viscosity of a protein solution with concentration $c$, $[\eta]$ the intrinsic viscosity of the protein, $k$ the self-crowding factor and $v$ the Einstein coefficient. For ideal hard-spheres $v = 2.5$, and for ideal spherical particles at infinite dilution, we obtain $k/v = 1/2.5 = 0.4$. Deviations of $k/v$ from this value are due to anisotropic protein shapes, including the effect of the hydration layer, or hydrodynamic interactions. Equation (3.3) can also be rewritten as

$$\frac{1}{D(c)} = \frac{1}{D_0} \exp\left(\frac{[\eta]c}{1 - k/vc[\eta]}\right), \tag{3.4}$$

with the Hb diffusion coefficients $D(c)$ and $D_0$ at concentration $c$, and at infinite dilution, respectively. Solving equation (3.4) for $c$ yields

$$c = \frac{\ln(D_0/D(c))}{\ln(D_0/D(c)) \cdot k/v \cdot [\eta] + [\eta]}. \tag{3.5}$$

Longeville & Stingaciu analysed experimental neutron spin-echo (NSE) data from concentrated Hb solutions and estimated $[\eta] = 2.94 \pm 0.79 \times 10^{-3}\,l\,g^{-1}$ and $k/v = 0.52 \pm 0.29$ [6]; we used these values for our QENS data analysis. These parameter values are specific for Hb, and allowed us to estimate the concentration of Hb when the diffusion coefficient is assumed to be known. Thus, following equation (3.5), we calculated theoretical values of Hb diffusion coefficients at infinite dilution $D_0$ by using the computer program HYDROPRO [58]. We estimated values of $D_0 = 46.8 \times 10^{-8}\,cm^2\,s^{-1}$ at 288 K, and $D_0 = 63.4 \times 10^{-8}\,cm^2\,s^{-1}$ at 298 K. The estimated value of the intrinsic viscosity using HYDROPRO was $3.60 \times 10^{-3}\,l\,g^{-1}$; thus, the estimate obtained by us agrees with Longeville and Stingaciu within experimental error. However, both parameters, $[\eta]$ and $k/v$, are strongly coupled in equations (3.4) and (3.5); and replacement of the measured $[\eta]$ value by the theoretical one would directly influence the estimated $k/v$ ratio. Therefore, we used the measured $[\eta]$ and $k/v$ as reported in [6].

The measured self-diffusion coefficients $D_{eff}$ obtained from the QENS data, reported in table 2, were used in equation (3.5) to calculate the concentration of Hb within the native, swollen and expanded RBCs. The $D_{eff}$ values were corrected for the contribution of rotational diffusion to obtain the value for centre-of-mass diffusion before incorporation into equation (3.5). The estimated Hb concentrations, $c$, are given in table 3. Standard errors in $c$ were obtained in the analysis, but they were estimated to be approximately 1–2%; this does not include error propagation of the assumed parameter values [$\eta$] and $k/v$ on the errors of $c_{QENS}$ determined by QENS. The latter are, however, certainly not smaller than those of the Hb concentrations $c_{SANS}$ determined from the SANS results.

Based on these calculations, we found that the intracellular Hb concentrations in human and horse RBCs, under physiological conditions, are $c_{QENS}$ 357 and 315 mg ml$^{-1}$, respectively. These estimates are close to each other with an average value of 336 mg ml$^{-1}$, which corresponds to an Hb volume fraction of $\phi = 0.25$. The $c_{QENS}$ values are within the errors the same as the intracellular Hb concentrations $c_{SANS}$ in human and horse RBCs of 332 and 341 mg ml$^{-1}$ obtained from SANS data

**Table 3.** Effective Hb concentrations in RBCs estimated from experimental QENS data. Coefficients of variation of the concentrations were between 1% and 2% based on the QENS errors.

| horse RBCs at $T = 288$ K | $c$ (mg ml$^{-1}$) | human Hb at $T = 298$ K | $c$ (mg ml$^{-1}$) |
|---|---|---|---|
| RBCs physiological state | 315 | RBCs physiological state | 357 |
| RBCs + valinomycin | 362 | RBCs + monensin | 295 |
| | | RBCs + monensin, glucose reduced metabolically | 297 |
| | | RBCs lysate | 281 |

(table 1 and figure 4). All $c_{QENS}$ and $c_{SANS}$ values are close to the value of 330 mg ml$^{-1}$ reported previously for human RBCs [2]. This comparison validates our experimental SANS and QENS results and the underlying theoretical concepts that we used and which was applied by Longeville & Stingaciu [6] for their NSE data.

Our study, therefore, is consistent with the interpretation of Longeville & Stingaciu [6] that Hb concentration in RBCs, under physiological conditions, has been selected by evolution to allow the optimal function of these cells, primarily for oxygen uptake in the lungs. Furthermore, it asserts that short-time self-diffusion of Hb in whole cells can be described quantitatively using suitable theoretical models that empirically consider protein anisotropy, the relevance of the hydration layer and hydrodynamic interactions.

Changes in Hb concentration are directly proportional to volume changes, when the number of molecules per cell remains the same; this was the case for our RBCs. Hence, based on our QENS data, addition of valinomycin resulted on an average on a reduction of RBC volume by 15%, while monensin led to RBC expansion by 17%. Those observations are fully supported by our USANS and SANS data (see §3.1): (i) USANS data show both swelling of RBCs upon addition of monensin in terms of an increased RBC–RBC correlation length, while valinomycin results in the shrinkage of RBC visible by the loss of that correlation length and an observable effective Guinier radius in the range of a few micrometres. (ii) RBC volume changes have also been observed by SANS, with an average RBC expansion/shrinkage of around 20%. Kinetic TOF SANS experiments validated the stability of those modified RBC shapes over at least 1 day. Observable RBC volume changes induced by the ionophores are more subtle (approx. 20% volume change) as those induced by high and low buffer osmolarity (approx. 30% volume change, table 1) [2]. Consumption of glucose by RBCs, and perturbation of the Na,K-ATPase did not further change RBC volume. This demonstrated that the concentrations of ionophore that we used were adequate to bring about major effects on RBC shape. Upon lysis, an even lower concentration of 281 mg ml$^{-1}$ was observed. If we consider an initial haematocrit value between 0.6 and 0.7, with an assumed human RBC concentration of 357 mg ml$^{-1}$, as estimated from the QENS data, then we would have expected an Hb concentration ranging from 214 to 250 mg ml$^{-1}$ after full lysis. The Hb concentration estimated from the RBC lysate agrees with that theoretically predicted, although it is on the upper limit. A simple explanation for the latter could be incomplete lysis after only one freeze–thawing cycle. This warrants further study in the future.

One important parameter that regulates protein diffusion at high concentrations is the concentration-dependent hydrodynamic function $H(c) = D(c)/D_0$, where $D(c)$ is the diffusion coefficient at concentration $c$, and $D_0$ is the corresponding value at infinite dilution [36]. It describes how global diffusion $D(c)$ of proteins slows down in response to the concentration $c$ of crowding particles. The effective slowing down depends on several parameters such as protein shape and its size compared to the crowding particles. Molecular motion in cells is responsible for a large variety of biological processes such as transport of RNA, DNA and proteins in cells. Hence, it is a topical field of research to study the dependence of $H(c)$ for different proteins in response to crowding [7,36,59]. For coherent X-ray or neutron scattering experiments, a $q$-dependence of $H(q,c)$ is seen [5]. For incoherent neutron scattering experiments, such as QENS, which directly probe particle self-diffusion, a $q$-dependence of $H$ at the probed large $q$-vectors is not expected. This is a major advantage of high-resolution QENS when compared with NSE or X-ray photon correlation spectroscopy (XPCS) experiments [36,60]. Techniques that allow us to determine $H(c)$ on the short-time limit are for example, neutron spectroscopy, while its long-time limit can be accessed by NMR, dynamic light scattering (DLS) or XPCS. By combining QENS and NMR data, the diffusion of Hb in crowded environments could be accessed on both the short- and long-time limit, which would provide an interesting perspective for future studies.

By using the theoretical diffusion coefficient at infinite dilution of $D_0 = 63.4 \times 10^{-8}$ cm$^2$ s$^{-1}$ at 298 K, we estimated a hydrodynamic value for short-time self-diffusion $H_{s,short} = 19.0 \times 10^{-8}$ cm$^2$ s$^{-1}$/1.27/ $63.4 \times 10^{-8}$ cm$^2$ s$^{-1}$ = 0.24 from the QENS data from the RBC lysate. Protein concentrations were approximately 280 mg ml$^{-1}$ for the QENS sample. This value corresponded to a volume fraction of $\phi = 0.20$. By considering the short-time range probed by QENS, our results indicated that hydrodynamic interactions of Hb in concentrated solutions were stronger than predicted from the theory for simple colloidal hard-sphere suspensions; this predicted a value of $H = 1 - 1.831\phi = 0.63$ for a volume fraction of $\phi = 0.20$ being equivalent to the QENS RBC lysate sample [61]. This result was in agreement with previous reports on globular, multi-domain and intrinsically disordered proteins [5,59,62–65].

The observed discrepancy between the experimentally measured hydrodynamic values for short-time diffusion, and the simplified theoretical hard-sphere case, was attributed to: (i) internal protein flexibility [62,63]; or (ii) the larger hydrodynamic radius of the protein due to the hydration layer [5,7,64,65]; or (iii) anisotropic shape [59]. Protein flexibility will play a minor role in the rather rigid and compact Hb molecules, and it will be more likely that their hydration, shape or charge lead to the observed deviation from the predicted behaviour, when the simple hard-sphere model is considered by comparison.

The properties describing particle shape and size are included empirically in $[\eta]$ and $k/v$ that were used in our analysis (above); they could be calculated from computer simulations (that are outside the scope of the present study). While this finding is consistent with an Hb molecule undergoing 'obstructed diffusion', it is not possible to interpret the results in terms of a simple hard-sphere model.

There is ambiguity in the literature on the physical nature of the interactions between Hb molecules in solution; and several dissimilar models have been used to interpret SANS data and diffusion measurements. For example, Han & Herzfeld [66] describe the concentration dependence of the diffusion coefficient with an excluded volume approach. Their work implies that the only interaction between particles is of the hard-sphere type in which longer range electrostatic effects, such as described by Krueger & Nossal [2] or Krueger et al. [4] as well as presented in the present work (above), are neglected.

# 4. Conclusion

In the present work, we analysed data from USANS/SANS and QENS experiments that showed the influence of RBC volume on Hb–Hb interactions, and on the internal dynamics and mean diffusion coefficient of Hb in horse and human RBCs. Morphological transitions of RBCs were induced by the ionophores monensin and valinomycin. Changes of RBC ultrastructure and modification of cellular packing arrangements were readily identified in the USANS data. The internal dynamics of Hb were largely independent of RBC shape, which agrees with the assumption that internal protein dynamics are decoupled from global protein diffusion. However, both local Hb–Hb interactions and global protein diffusion depended strongly on the volume of the RBC, being sensitive indicators of local protein concentration. Intracellular Hb concentration was able to be calculated from the measured Hb–Hb structure factors, treating the proteins as charged hard-spheres. Using an analytical model that contains parameters that account for Hb shape, hydration and hydrodynamics, we estimated the protein concentration in RBCs based on dynamic QENS data, and studied the effect of morphological changes of the cells on the intracellular Hb concentration. Both neutron-beam approaches yielded the same intracellular Hb concentrations within the accuracy of these techniques and demonstrated that both ionophores used modified RBC volume by approximately 20%. Its shape, hydration and charge affect Hb's diffusion coefficient when estimated on the nanometre-to-nanosecond length and time ranges, when compared directly with theoretical values of hard-sphere colloids at comparable volume fractions.

Data accessibility. Experimental USANS/ SANS and QENS data presented in this manuscript are available as electronic supporting material (https://doi.org/10.6084/m9.figshare.8798720.v3).
Authors' contributions. K.S., N.R.d.S., L.d.C., A.E.W., P.W.K., C.J.G. and A.M.S. performed neutron scattering experiments and sample preparation laboratory work; M.S. and A.M.S. carried out the data analyses; A.M.S. and C.J.G. designed and coordinated the study; the manuscript was written by A.M.S., C.J.G. and P.W.K. with input from all authors. All authors gave final approval for publication.
Competing interests. We declare we have no competing interest.

Funding. The work was supported by a DFG grant no. STA 1325/2-1 to A.M.S. M.S. acknowledges the support of the International Helmholtz Research School of Biophysics and Soft Matter (BioSoft).

Acknowledgements. The experiments were performed at the ANSTO instruments KOOKABURRA, QUOKKA, BILBY and EMU at, Lucas Heights, New South Wales, Australia. We thank ANSTO for provision of neutron beam time.

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
