## [Reviewer comments · Royal Society Open Science]

Review History

RSOS-200391.R0 (Original submission)

Review form: Reviewer 1

Is the manuscript scientifically sound in its present form?

No

Are the interpretations and conclusions justified by the results?

No

Is the language acceptable?

No

Do you have any ethical concerns with this paper?

No

Have you any concerns about statistical analyses in this paper?

No

Recommendation?

Major revision is needed (please make suggestions in comments)

Comments to the Author(s)

After reading the revised version of the manuscript I have more comments.

In the abstract, the authors mention the presence of Hb₄ tetramers, the formation of condensed protein phases or Hb assemblies.

The authors conclude: "The formation of large-scale structures in the RBC lysate might be detectable in the reduced long-time diffusion coefficient seen by NMR as compared to the short-time value that is probed by QENS. Hence, our observations could tentatively be explained by the presence of large scale clusters or aggregates of Hb in the RBC lysate".

From the abstract and the conclusion one deals with a vague information about the large scale structure of RBC lysate.

More, it looks as NMR and QENS techniques are revealing large scale clusters in the RBC lysate. It is not the more direct way to access the large scale structure of a system. Small angle neutron (or x-ray) scattering) is more appropriate.

The strategy adopted by the authors is not logical. They propose a description of the dynamics of RBC lysate on the basis of assumptions of the large scale structure while they have SANS results.

I suggest that the authors include in their manuscript the SANS results or if not, they publish them separately. But in the latter case the manuscript has to be postponed.

Review form: Reviewer 2

Is the manuscript scientifically sound in its present form?

No

Are the interpretations and conclusions justified by the results?

No

Is the language acceptable?

Yes

Do you have any ethical concerns with this paper?

No

Have you any concerns about statistical analyses in this paper?

Yes

Recommendation?

Reject

Comments to the Author(s)

I happened to review the original submission to the manuscript to the Journal of the Royal Society Interface. Reviewing the other comments, I find that there are several valid comments from other reviewers as well. In the revised submission of the manuscript to the Royal Society Open Science, it is rather unfortunate that the authors have not addressed many of the significant issues in the manuscript. Though the authors tend to agree with the comments, the corresponding revision in the manuscript is missing. The overall structure and content of the original submission are intact with some edits to the text.

The main concerns from the reviews are not addressed. In addition there are several apparent errors, and some of them are listed below.

(a) It is rather absurd to arbitrary reference the water to 3.9 ppm (albeit it does not change the diffusion coefficient) and highlight it with a spectrum and a separate table listing the chemical shifts.

- (b) How can figure 3 has a linear dependence when the x-axis is the magnetic field gradient strength? Figures 4 and 5 (Stejskal-Tanner plot) also shows an expected linear trend. The latter is an area of the gradient pulse.
- (c) What is the origin of error bars in Figure 6? What is the uncertainty in comparing two different experiments?
- (d) How can the ^1H NMR frequency be at 400.13 MHz in an AMX 300 wide bore spectrometer?

Decision letter (RSOS-200391.R0)

Dear Dr Stadler:

Manuscript ID RSOS-200391 entitled "Haemoglobin Dynamics in Concentrated Solutions and Red Blood Cells: Neutron Spectroscopy and Magnetic Field Gradient NMR Studies" which you submitted to Royal Society Open Science, has been reviewed. The comments from reviewers are included at the bottom of this letter.

In view of the criticisms of the reviewers, the manuscript has been rejected in its current form. However, a new manuscript may be submitted which takes into consideration these comments.

Please note that resubmitting your manuscript does not guarantee eventual acceptance, and that your resubmission will be subject to peer review before a decision is made.

Your resubmitted manuscript should be submitted by 25-Oct-2020. If you are unable to submit by this date please contact the Editorial Office.

on behalf of Professor Luning Liu (Associate Editor) and Pietro Cicuta (Subject Editor)
openscience@royalsociety.org

Associate Editor Comments to Author (Professor Luning Liu):
Comments to the Author:

Thank you for submitting your manuscript to Royal Society Open Science. We have obtained constructive comments from two reviewers, who have raised critical concerns about the experimental design, technique, as well as data analysis and interpretation. Based on their advice and the fact that the manuscript still needs considerable improvements, I am afraid that I cannot accept it for publication in Royal Society Open Science.

Reviewers' Comments to Author:
Reviewer: 1

Comments to the Author(s)

After reading the revised version of the manuscript I have more comments.

In the abstract, the authors mention the presence of Hb₄ tetramers, the formation of condensed protein phases or Hb assemblies.

The authors conclude: "The formation of large-scale structures in the RBC lysate might be detectable in the reduced long-time diffusion coefficient seen by NMR as compared to the short-time value that is probed by QENS. Hence, our observations could tentatively be explained by the presence of large scale clusters or aggregates of Hb in the RBC lysate".

From the abstract and the conclusion one deals with a vague information about the large scale structure of RBC lysate.

More, it looks as NMR and QENS techniques are revealing large scale clusters in the RBC lysate. It is not the more direct way to access the large scale structure of a system. Small angle neutron (or x-ray) scattering) is more appropriate.

The strategy adopted by the authors is not logical. They propose a description of the dynamics of RBC lysate on the basis of assumptions of the large scale structure while they have SANS results. I suggest that the authors include in their manuscript the SANS results or if not, they publish them separately. But in the latter case the manuscript has to be postponed.

Reviewer: 2

Comments to the Author(s)

I happened to review the original submission to the manuscript to the Journal of the Royal Society Interface. Reviewing the other comments, I find that there are several valid comments from other reviewers as well. In the revised submission of the manuscript to the Royal Society Open Science, it is rather unfortunate that the authors have not addressed many of the significant issues in the manuscript. Though the authors tend to agree with the comments, the corresponding revision in the manuscript is missing. The overall structure and content of the original submission are intact with some edits to the text.

The main concerns are from the reviews are not addressed. In addition there are several apparent errors, and some of them are listed below.

(a) It is rather absurd to arbitrary reference the water to 3.9 ppm (albeit it does not change the diffusion coefficient) and highlight it with a spectrum and a separate table listing the chemical shifts.

(b) How can figure 3 has a linear dependence when the x-axis is the magnetic field gradient strength? Figures 4 and 5 (Stejskal-Tanner plot) also shows an expected linear trend. The latter is an area of the gradient pulse.

(c) What is the origin of error bars in Figure 6? What is the uncertainty in comparing two different experiments?

(d) How can the ¹H NMR frequency be at 400.13 MHz in an AMX 300 wide bore spectrometer?

Author's Response to Decision Letter for (RSOS-200391.R0)

See Appendix A.

RSOS-201507.R0

Review form: Reviewer 1

Is the manuscript scientifically sound in its present form?

Yes

Are the interpretations and conclusions justified by the results?

Yes

Is the language acceptable?

Yes

Do you have any ethical concerns with this paper?

No

Have you any concerns about statistical analyses in this paper?

No

Recommendation?

Accept with minor revision (please list in comments)

Comments to the Author(s)

The authors have changed the title of the manuscript that is now more appropriate since they have included in their manuscript the results of USANS and SANS about the RBCs. I have minor questions:

Abstract: Line 32, the authors wrote: "Quasielastic neutron scattering (QENS) experiments were performed to measure diffusive motion of Hb in RBCs and in concentrated Hb solution". Why do they mention: concentrated Hb solution? The authors must precisely mention that their sample for USANS, SANS and QENS deals with RBCs suspensions.

The following captions have to be completed: "RBC physiological" instead of "RBC" for Fig.1, Fig. 6A, Fig.7B, Fig. 8 A and B, Fig.9A.

Decision letter (RSOS-201507.R0)

Dear Dr Stadler,

On behalf of the Editors, we are pleased to inform you that your Manuscript RSOS-201507 "Effect of Red Blood Cell Shape Changes on Haemoglobin Interactions and Dynamics: A Neutron

Scattering Study" has been accepted for publication in Royal Society Open Science subject to minor revision in accordance with the referees' reports. Please find the referees' comments along with any feedback from the Editors below my signature.

Please submit your revised manuscript and required files (see below) no later than 7 days from today's (ie 10-Sep-2020) date. Note: the ScholarOne system will 'lock' if submission of the revision is attempted 7 or more days after the deadline. If you do not think you will be able to meet this deadline please contact the editorial office immediately.

When resubmitting your manuscript, please ensure you check and update the following email address for one of your co-authors, which is currently marked as 'invalid' by our system:

m.sarter@fz-juelich.de

Best regards,

on behalf of Professor Luning Liu (Associate Editor) and Pietro Cicuta (Subject Editor)
openscience@royalsociety.org

Associate Editor Comments to Author (Professor Luning Liu):

The revised manuscript has been reviewed by the reviewers. They are basically happy with what have been modified and the suppression of NMR results. Based on their comments, I am pleased to recommend an acceptance of the manuscript with minor revisions as suggested by the reviewer (see below).

Reviewer comments to Author:

Reviewer: 1
Comments to the Author(s)

The authors have changed the title of the manuscript that is now more appropriate since they have included in their manuscript the results of USANS and SANS about the RBCs. I have minor questions:

Abstract: Line 32, the authors wrote: “Quasielastic neutron scattering (QENS) experiments were performed to measure diffusive motion of Hb in RBCs and in concentrated Hb solution”. Why do they mention: concentrated Hb solution? The authors must precisely mention that their sample for USANS, SANS and QENS deals with RBCs suspensions.

The following captions have to be completed: “RBC physiological” instead of “RBC” for Fig.1, Fig. 6A, Fig.7B, Fig. 8 A and B, Fig.9A.

===PREPARING YOUR MANUSCRIPT===

===PREPARING YOUR REVISION IN SCHOLARONE===

Author's Response to Decision Letter for (RSOS-201507.R0)

See Appendix B.

Decision letter (RSOS-201507.R1)

Dear Dr Stadler,

It is a pleasure to accept your manuscript entitled "Effect of Red Blood Cell Shape Changes on Haemoglobin Interactions and Dynamics: A Neutron Scattering Study" in its current form for publication in Royal Society Open Science. The comments of the reviewer(s) who reviewed your manuscript are included at the foot of this letter.

on behalf of Professor Luning Liu (Associate Editor) and Pietro Cicuta (Subject Editor)
openscience@royalsociety.org

Appendix A

Reviewer: 1

Comments to the Author(s)

After reading the revised version of the manuscript I have more comments.

Comment 1:

In the abstract, the authors mention the presence of Hb₄ tetramers, the formation of condensed protein phases or Hb assemblies.

The authors conclude: "The formation of large-scale structures in the RBC lysate might be detectable in the reduced long-time diffusion coefficient seen by NMR as compared to the short-time value that is probed by QENS. Hence, our observations could tentatively be explained by the presence of large scale clusters or aggregates of Hb in the RBC lysate".

From the abstract and the conclusion one deals with a vague information about the large scale structure of RBC lysate.

Authors' reply to comment 1:

Following the comments of referee 2 we have decided to delete the entire NMR data set and all conclusions that have been based on NMR data. Hence, the mentioned statements in the abstract as well as the part of the discussion concerning the large-scale structure in the RBC lysate have been deleted from the manuscript as they had been based on NMR data.

Please note, however, that SANS experiments performed on RBC samples provided strong experimental evidence for a mass-fractal like structure in RBC that can be attributed to a large-scale structure or fluid condensed phases of Hb as we had written in the previous version of the manuscript (see Figure 2 and page 16, second paragraph, describing SANS data in the revised version of the manuscript).

Please note as well that we have deleted the words 'tetrameric haemoglobin' and 'Hb tetramers' in the abstract and replaced them by 'haemoglobin' and 'Hb' as they appear to be confusing. We just wanted to say that haemoglobin is a homotetramer consisting of two alpha and two beta chains (see page 3, second sentence in 1. Introduction).

Changes to the manuscript:

NMR data and all conclusions that were drawn from NMR data have been deleted from the manuscript. Additionally, the word 'tetramer' has been deleted from the abstract for clarity.

Comment 2:

More, it looks as NMR and QENS techniques are revealing large scale clusters in the RBC lysate. It is not the more direct way to access the large scale structure of a system. Small angle neutron (or x-ray) scattering) is more appropriate.

The strategy adopted by the authors is not logical. They propose a description of the dynamics of RBC lysate on the basis of assumptions of the large scale structure while they have SANS results.

I suggest that the authors include in their manuscript the SANS results or if not, they publish them separately. But in the latter case the manuscript has to be postponed.

Authors' reply to comment 2:

We agree with the comment of referee 1. We have analyzed in the meanwhile an extensive set of ultra-small-angle neutron scattering (USANS) and small-angle neutron scattering (SANS) data that we had measured on the instruments KOOKABURRA, QUOKKA and BILBY at ANSTO. Initially, we had planned to publish the USANS/ SANS data separately. However, we have decided to include the USANS/ SANS data set in the revised version of the manuscript (Figures 1, 2, 3, 4, 5 and Table 1) as suggested by referee 1. We added a description of the experimental USANS/ SANS studies to Materials and Methods (2.2 USANS and SANS experiments) and discuss the USANS/ SANS results in section 3.1 RBC morphology and Hb-Hb interactions investigated by USANS/ SANS.

The rationale to include the USANS/ SANS data into the manuscript is the following: The results obtained from the USANS/ SANS experiments and the dynamics data obtained from our QENS experiments really fit excellently together and strongly support each other. Hence, we have decided to include the USANS/ SANS data in the revised version of our manuscript currently in review at *Royal Society Open Science* instead of publishing the USANS/ SANS data separately. A complete and coherent picture on the behavior of Hb in RBCs emerges that is strongly supported by the combined structural and dynamics data.

Changes to the manuscript:

We have added the USANS/ SANS data to the manuscript and discuss these data in the revised version of the manuscript. We again would like to highlight here that our previous results based on QENS experiments are fully supported by the USANS/ SANS data and have remain fully unchanged throughout the entire revision.

Reviewer: 2

Comments to the Author(s)

I happened to review the original submission to the manuscript to the Journal of the Royal Society Interface. Reviewing the other comments, I find that there are several valid comments from other reviewers as well. In the revised submission of the manuscript to the Royal Society Open Science, it is rather unfortunate that the authors have not addressed many of the significant issues in the manuscript. Though the authors tend to agree with the comments, the corresponding revision in the manuscript is missing. The overall structure and content of the original submission are intact with some edits to the text.

The main concerns are from the reviews are not addressed. In addition there are several apparent errors, and some of them are listed below.

Comments from referee 2:

- (a) It is rather absurd to arbitrary reference the water to 3.9 ppm (albeit it does not change the diffusion coefficient) and highlight it with a spectrum and a separate table listing the chemical shifts.
- (b) How can figure 3 has a linear dependence when the x-axis is the magnetic field gradient strength? Figures 4 and 5 (Stejskal-Tanner plot) also shows an expected linear trend. The latter is an area of the gradient pulse.
- (c) What is the origin of error bars in Figure 6? What is the uncertainty in comparing two different experiments?
- (d) How can the ^1H NMR frequency be at 400.13 MHz in an AMX 300 wide bore spectrometer?

Authors reply to comments from referee 2:

We have deleted the NMR data from the manuscript and all conclusions that have been based on the NMR data. Hence, comments of referee 2 are no longer relevant for the revised version of our manuscript.

Changes to the manuscript:

The entire NMR data has been removed from the manuscript (Figures 1, 2, 3, 4, 5, 6 and Tables 1 from the previous version of the manuscript) and replaced by USANS/ SANS data. All conclusions that were drawn from NMR data have been deleted from the revised version of the manuscript.

Results obtained by QENS remain fully unchanged and are strongly supported by new USANS/ SANS data.

Appendix B

Associate Editor Comments to Author (Professor Luning Liu):

The revised manuscript has been reviewed by the reviewers. They are basically happy with what have been modified and the suppression of NMR results. Based on their comments, I am pleased to recommend an acceptance of the manuscript with minor revisions as suggested by the reviewer (see below).

Reviewer comments to Author:

Reviewer: 1

Comments to the Author(s)

The authors have changed the title of the manuscript that is now more appropriate since they have included in their manuscript the results of USANS and SANS about the RBCs. I have minor questions:

Abstract: Line 32, the authors wrote: “Quasielastic neutron scattering (QENS) experiments were performed to measure diffusive motion of Hb in RBCs and in concentrated Hb solution”. Why do they mention: concentrated Hb solution? The authors must precisely mention that their sample for USANS, SANS and QENS deals with RBCs suspensions.

Authors' reply:

We have measured several red blood samples (RBC) including one RBC lysate with QENS. The RBC lysate is a highly concentrated haemoglobin (Hb) solution. For clarity we replaced concentrated 'Hb solution' in the abstract by 'an RBC lysate'.

The following captions have to be completed: “RBC physiological” instead of “RBC” for Fig.1, Fig. 6A, Fig.7B, Fig. 8 A and B, Fig.9A.

Authors' reply:

Thank you indeed for finding that mistake. We replaced the word 'RBC' by 'RBC physiological' in the figure legends of Figs 1, 6, 7, 8 and 9 to be compatible with the other figures showing SANS data (Figs. 2, 3, 4, 5).